



# Analysis of New Particle Formation (NPF) Events at Nearby Rural, Urban Background and Urban Roadside Sites

**Dimitrios Bousiotis[1], Manuel Dall'Osto[2], David C.S. Beddows[1], Francis D. Pope[1] and Roy M. Harrison[1a*]**

[1] School of Geography, Earth & Environmental Sciences and
National Centre for Atmospheric Science
University of Birmingham, Edgbaston, Birmingham
B15 2TT, United Kingdom

[2] Institute of Marine Sciences, CSIC
Passeig Marítim de la Barceloneta, 37-49. E-08003
Barcelona, Spain

[a]Also at: Department of Environmental Sciences / Center of Excellence in Environmental Studies, King Abdulaziz University, PO Box 80203, Jeddah, 21589, Saudi Arabia

* To whom correspondence should be addressed.
Tele: +44 121 414 3494; Fax: +44 121 414 3709; Email: r.m.harrison@bham.ac.uk



**ABSTRACT**
NPF events have different patterns of development depending on the conditions of the area in which
they occur. In this study, NPF events occurring at three sites of differing characteristics (rural
Harwell (HAR), urban background North Kensington (NK), urban roadside Marylebone Road
(MR), London, UK) were studied (seven years of data). The different atmospheric conditions in
each study area not only have an effect on the frequency of the events, but also affect their
development. The frequency of NPF events is similar at the rural and urban background locations
(about 7% of days), with a high proportion of events occurring at both sites on the same day (45%).
The frequency of NPF events at the urban roadside site is slightly less (6% of days), and higher
particle growth rates (average 5.5 nm h$^{-1}$ at MR compared to 3.4 nm h$^{-1}$ and 4.2 nm h$^{-1}$ at HAR and
NK respectively) must result from rapid gas to particle conversion of traffic-generated pollutants.
A general pattern is found in which the condensation sink increases with the degree of pollution of
the site, but this is counteracted by increased particle growth rates at the more polluted location.  A
key finding of this study is that the role of the urban environment leads to an increment of 20% in
$N_{16\text{-}20nm}$ in the urban background compared to that of the rural area in NPF events occurring at both
sites. The relationship of the origin of incoming air masses is also considered and an association of
regional events with cleaner air masses is found. Due to lower availability of condensable species,
NPF events that are associated with cleaner atmospheric conditions have lower growth rates of the
newly formed particles. The decisive effect of the condensation sink in the development of NPF
events and the survivability of the newly formed particles is underlined, and influences the overall



contribution of NPF events to the number of ultrafine particles in an area. The other key factor
identified by this study is the important role that urban pollution plays in new particle formation
events.





## 1.   INTRODUCTION

Ultrafine particles (particles with diameter smaller than 100 nm) typically make the greatest contribution in the total particle count, especially in urban environments (Németh et al., 2018), but a very small contribution to total volume and mass (Harrison et al., 2000). Research studies have indicated that ultrafine particles can cause pulmonary inflammation and may contribute to cardiovascular disease (Oberdörster, 2000) and have increased possibility to penetrate the brain and central nervous system (Politis et al., 2008) compared to fine and coarser particles. Since some studies report that toxicity per unit mass increases as particle size decreases (Penttinen et al., 2001; MacNee et al., 2003; Davidson et al., 2005); it is considered possible that particle number concentrations may be a better predictor of health effects than mass concentrations (Harrison et al., 2000; Atkinson et al., 2010; Kelly et al., 2012; Samoli et al., 2016). Additionally, NPF events have an impact on climate (Makkonen et al., 2012) either by increasing the number of cloud condensation nuclei (Spracklen et al., 2008; Merikanto et al., 2009; Dameto de España et al., 2017; Kalkavouras et al., 2017), or directly affecting the optical properties of the atmosphere (Seinfeld et al., 2012).

The sources of ultrafine particles in urban areas can either be primary emissions from traffic (Shi et al., 1999; Harrison et al., 2000), airports (Masiol et al., 2017) and other combustion related processes (Keuken et al., 2015; Kecorius et al., 2016), or by new particle formation (NPF) from gaseous precursors. NPF as described by Kulmala et al. (2014), is the process of production of low-



volatility vapours, clustering of these vapours, nucleation, activation of the clusters with a second
group of vapours and condensational growth to larger sizes. This process can occur both locally or
on a larger scale; in the latter case the events are characterized as regional. Regional events have
been found to take place in a scale of hundreds of kilometres (Németh et al., 2014; Shen et al.,
2018), without being affected by air mass advection (Salma et al., 2016). NPF is one of the main
contributors of particles in the atmosphere (Spracklen et al., 2010; Kulmala et al., 2016; Rahman et
al., 2017) and this contribution increases moving from a kerbside to a rural area (Ma et al., 2015).
While NPF events in rural and remote areas have been widely studied for many years (O'Dowd et
al., 2002; Dal Maso et al., 2005; Ehn et al., 2010; Dall'Osto et al., 2017; Kalkavouras et al., 2017),
in urban areas intensive studies have started mainly in recent years (Jeong et al., 2010; Minguillón
et al., 2015; Peng et al., 2017; Németh et al., 2018). Early studies in Birmingham, UK highlighted
the connection of NPF events with solar radiation (Shi et al., 2001) and a low condensation sink
(Alam et al., 2003), a measure of pre-existing aerosol loading (Dal Maso et al., 2002). The
importance of a low condensation sink was further underlined by later studies, as being one of the
most influential variables in the occurrence of NPF in all types of environment (Wehner et al.,
2007; Park et al., 2015; Pikridas et al., 2015). An important contributor to many NPF pathways is
$SO_2$ (Woo et al., 2001; Berndt et al., 2006; Laaksonen et al., 2008), which in the presence of solar
radiation forms $H_2SO_4$, often the main component of the initial clusters (Kuang et al., 2008;
Kulmala et al., 2013; Bianchi et al., 2016; Kirkby et al., 2016). Dall'Osto et al. (2013) pointed out
that the role of $SO_2$ is less significant in urban areas compared to rural and background areas. $SO_2$



concentration variability in urban areas was found to have a small impact on the frequency of NPF
events (Alam et al., 2003; Jeong et al., 2010), though it can have an effect on the number of
particles formed (Charron et al., 2007). Furthermore, Dall'Osto et al. (2018) in their research at 24
sites in Europe, pointed out the different role $SO_2$ seems to play depending on its concentration, and
that of other species. Jayaratne et al. (2017) however found that in the heavily polluted environment
of Beijing, China, NPF events were more probable in sulphur rich conditions rather than sulphur
poor. Apart from its role in the initial formation of the clusters, $H_2SO_4$ seems to participate in the
early stages of growth of the newly formed clusters (Kulmala et al., 2005; Iida et al., 2008; Xiao et
al., 2015). In later stages of growth, low or extremely low volatility organic compounds (O'Dowd et
al., 2002; Laaksonen et al., 2008; Metzger et al., 2010; Kulmala et al., 2013; Tröstl et al., 2016;
Dall'Osto et al., 2018) appear to be more important, while the role of ammonium nitrate in particle
growth is also considered (Zhang et al., 2017). While in rural areas the organic compounds are
mainly of biogenic origin (Riccobono et al., 2014; Kirkby et al., 2016), in urban areas they mainly
originate from combustion procedures (Robinson et al., 2007; Gentner et al., 2012). Many
comparative studies have reported higher growth rates in urban areas compared to background  sites
(Wehner et al., 2007; Jeong et al., 2010; Salma, et al., 2016; Wang et al., 2017), as well as greater
particle formation rates (Salma, et al., 2016; Nieminen et al., 2018) and a higher frequency of NPF
events (Peng et al., 2017), which was attributed to the higher concentration of condensable species.
Salma et al. (2014) however reported fewer NPF events in the city centre of Budapest compared to
the urban background, due to the higher condensation sink. Due to the complexity of the conditions





and mechanisms within an urban area (Harrison, 2017), NPF events are harder to study and factors

to be attributed. A large number of particles of size 1.3 – 3 nm has been attributed to traffic

emissions at a kerbside site and thus not related to homogeneous nucleation mechanisms (Rönkkö et

al., 2017; Hietikko et al., 2018) and studies in Barcelona, Spain (Dall'Osto et al., 2012; Brines et al.,

2014) and Leicester, U.K. (Hama et al., 2017), attributed a larger portion of nucleation mode

particles to vehicular emissions compared to photochemically induced nucleation. As the

condensation sink is higher within an urban environment, NPF events are less favoured. Their

occurrence is attributed to either ineffective scavenging or the higher growth rate of the newly

formed particles (Kulmala et al., 2017), when sufficient concentrations of precursors are present in

the atmosphere (Fiedler et al., 2005), as particle formation was found to take place on both event

and non-event days (Riipinen et al., 2007).

In this study, NPF events in three areas of different land use in the southern U.K. are analyzed.

Studies for NPF events have been conducted in the past for Harwell, Oxfordshire (Charron et al.,

2007; 2008) and the effect of NPF upon particle size distributions was also considered for N.

Kensington, London (Beddows et al., 2015). A combined study including all three sites has also

been conducted, but in the aspect of ultrafine particle variation (Von Bismarck-Osten et al., 2013).

The present study is the first to use a combined long term database for all three sites, focusing on

the trends and conditions of NPF events at these sites, as well as the first which identifies NPF

events at the highly trafficked Marylebone Road site, as up to this point ultrafine particles were





attributed only to traffic (Charron et al., 2003; Dall'Osto et al., 2011). As in this study a rural and an
urban background area are studied alongside a kerbside site in the city of London in close
proximity, the conditions and development of NPF events in a mid-latitude European region are
discussed in relation to the influence of different local environments.

**2.        DATA AND METHODS**
**2.1       Site Description and Data Availability**
This study analyses NPF events in three areas in the southern United Kingdom (Fig. 1). Harwell in
Oxfordshire, is located about 80 km west of the greater London area. The site is in the grounds of
the Harwell Science Centre in Oxfordshire (51° 34' 15" N, 1° 19' 31" W) and is representative of a
rural background area; a detailed description of the site was given by Charron et al. (2013). North
Kensington is a suburban area in the western side of London, U.K. The site is located in the grounds
of Sion Manning School (51° 31' 15" N, 0° 12' 48" W) and is representative of the urban
background of London. A detailed description of the site was given by Bigi and Harrison (2010).
Marylebone Road is located in the centre of London, U.K. The site is located on the kerbside of
Marylebone road (51° 31' 21" N; 0° 9' 16" W), a very busy arterial route within a street canyon. A
more detailed description of the area can be found in Charron and Harrison (2003).

At all three sites, seven years of particle number size distributions in the range of 16.6 – 604 nm
have been measured and recorded as 15-minute averages, using a Scanning Mobility Particle Sizer



(SMPS), comprised by an Electrostatic Classifier (EC, TSI model 3080) and a condensation Particle
Counter (CPC, TSI Model 3775), operated on behalf of the Department for Environment, Food and
Rural Affairs (DEFRA) in the U.K. At all sites the inlet air is dried, and operation is in accord with
the EUSAAR/ACTRIS protocol (Wiedensohler et al., 2012). These 15-minute measurements were
averaged to an hourly resolution. In Harwell there were 46930 hours of available SMPS data
(76.5% coverage), in N. Kensington 51059 (83.3% coverage) and in Marylebone 45562 (74.3%
coverage). A free-standing CPC (TSI model 3022A) also operated alongside for most of the years
of the survey and was used to give an estimate of particles in the 7-16.6 nm range by difference
from the SMPS.

Additionally, air pollutants and other aerosol chemical composition data were extracted from the
DEFRA website (https://uk-air.defra.gov.uk/). Meteorological data for Harwell and Heathrow
airport (used for N. Kensington and Marylebone road) were available from the Met Office, while
solar radiation data from Benson station (for Harwell) and Heathrow airport (for N. Kensington and
Marylebone Road), were extracted from the Centre for Environmental Data Analysis (CEDA) site
(http://www.ceda.ac.uk). Back trajectory data calculated using the HYSPLIT model (Draxler et al.,
1998),        were        extracted        by        the        NOAA        Air        Resources        Laboratory
(https://ready.arl.noaa.gov/READYtransp.php) and were processed using the Openair package for R
(Carslaw et al., 2012).





## 2.2    Methods

### 2.2.1    NPF events selection

The identification of the NPF event days was made by visual inspection of SMPS and CPC data

using the criteria set by Dal Maso et al. (2005). NPF events are considered when a distinctly new

mode of particles which appears in the size distribution at nucleation mode size, prevails for some

hours and shows signs of growth. Using these criteria, NPF events are classified into two classes, I

and II depending on the confidence level. Class I events are further classified to Ia and Ib, with class

Ia containing very clear and strong particle formation events, while Ib contains less clear events. In

this study the events of class Ia are only considered as being the most suitable for analysing case

studies of NPF events.   This analysis took account of the fact that nanoparticle emissions from

Heathrow Airport affect size distributions at London sites (Harrison et al., 2018), and such primary

emission influences were not included as NPF events.

### 2.2.2   Calculation of the condensation sink and growth rate

For the calculation of the condensation sink the method proposed in Kulmala et al. (2001) was used

in which the condensation sink is calculated as

$$CS = 4\pi D \sum \beta_M \, r \, N$$

(1)



where r is the radius of the particles and N is the number concentration of the particles. D is the
diffusion coefficient calculated (for T = 293 K and P = 1013.25 mbar) according to Polling et al.

195 (2000):


$$D_{vap} = 0.00143 \cdot T^{1}.75 \; \frac{\sqrt{M_{air}^{-1} + M_{vap}^{-1}}}{P\left(D_{x,air}^{\frac{1}{3}} + D_{x,vap}^{\frac{1}{3}}\right)^{2}}$$

197 (2)


where P is air pressure, M is the molar mass and $D_x$ is the diffusion volume for air and $H_2SO_4$. $\beta_M$ is
the Fuchs correction factor calculated as (Fuchs et al., 1971):

$$\beta_M = \frac{1 + K_n}{1 + \left(\frac{4}{3a} + 0.377\right)K_n + \frac{4}{3a}K_n^{2}}$$

202 (3)


where $K_n$ is the relation of the particle diameter and the mean free path of the gas $\lambda_m$, called the
Knudsen number.

The growth rate of the particles on nucleation event days was also calculated as proposed by
Kulmala et al. 2012, using the formula





$$GR = \frac{D_{P_2} - D_{P_1}}{t_2 - t_1}$$
(4)


for the period of each event day when growth was observed.

**2.2.3    Calculation of the urban increment (U.I.)**
The urban increment is defined as the ratio of the number concentration of particles below 20 nm
for event days to the average (for the period April – October, when the majority of the events take
place) for North Kensington to that at Harwell. This provides with a measure of the new particles
formed in each area in comparison to the average conditions, and is calculated by

$$U.I. = \frac{NK_{Nuc\,Max} - NK_{Bg}}{HW_{Nuc\,Max} - HW_{Bg}}$$
(5)


where $NK_{Nuc\,Max}$ is the maximum concentration of particles below 20 nm found in the diurnal cycle
on event days (found at 13:00) and $NK_{Bg}$ is the average mean concentration at the same time (same
for Harwell in the denominator).




10.5194/acp-2018-1057
Atmos. Chem. Phys. Discuss.



### 2.2.4 Calculation of nucleation strength factor (NSF) and the P parameter

The Nucleation Strength Factor (NSF) was proposed as a measure of the effect nucleation events have in the composition of ultrafine particles in an area. Two factors were proposed. First is the $NSF_{NUC}$. This is calculated as

$$NSF_{NUC} = \frac{\left(\frac{N_{(smallest\ size\ available-100)}}{N_{(100-largest\ size\ available)}}\right)_{nucleation\ days}}{\left(\frac{N_{(smallest\ size\ available-100)}}{N_{(100-largest\ size\ available)}}\right)_{non-nucleation\ days}} \qquad (6)$$

and provides of a measure of the concentration increment on nucleation days exclusively caused by new particle formation (NPF). The second factor is $NSF_{GEN}$ calculated as

$$NSF_{GEN} = \frac{\left(\frac{N_{smallest\ size\ available-100}}{N_{100-largest\ size\ available}}\right)_{all\ days}}{\left(\frac{N_{smallest\ size\ available-100}}{N_{100-largest\ size\ available}}\right)_{non-nucleation\ days}} \qquad (7)$$

and gives a measure of the overall contribution of NPF on a longer span (Salma et al. 2017).

The dimensionless survival parameter P, as proposed in Kulmala et al. (2017), was calculated as

$$P = \frac{CS'}{GR'}$$



where CS' = CS/($10^{-4}$ s$^{-1}$) and GR' = GR/(1 nm hour$^{-1}$). An increased P parameter is an indication
that a smaller percentage of newly formed particles will survive to greater sizes. Hence this is the
inverse of particle survivability, and values of P<50 are typically required for NPF in clean or
moderately polluted environments, although higher values of P are observed in highly polluted
atmospheres (Kulmala et al, 2017).

**3.        RESULTS AND DISCUSSION**
**3.1        NPF Events at the Background Areas**
**3.1.1        Conditions and trends of NPF events**
The number of NPF event days for each site per year, those that took place simultaneously on both
urban and rural background sites, as well as those events that took place in all three sites
simultaneously appear in Table 1. Given that overall data recovery was in the range of 74-83%,
results from individual years are unreliable, but the seven-year runs should average out most of the
effects of incomplete data recovery. The number of events is similar for Harwell and N.
Kensington, with a frequency of about 7% of all days with data. There is a clear seasonal variation
favouring summer and spring (Figure 2) for both areas of the study. A similar pattern of variation
was found for N. Kensington by Beddows et al. (2015). In general, higher solar radiation, lower
relative humidity, low cloud cover and higher pressure conditions, lower concentrations of
pollutants as well as lower condensation sink are found when NPF events took place in all areas
(Figure S1), as was also reported by Charron et al. (2007) for Harwell. While $SO_2$ is one of the main



factors for NPF events to occur, concentrations are lower when events take place. This is indicative
that $SO_2$ concentrations in these areas are sufficient for events to take place, and higher
concentrations are likely to be associated with higher pollution and a higher condensation sink. This
is also the case for gaseous ammonia (results not included) for Harwell where data was available, as
there was no distinct variation found between event and non-event days, but as the concentration of
ammonia in the U.K. is in the range of few ppb (Sutton et al., 1995), it is sufficient according to
ternary nucleation theory (Korhonen et al., 1999) for NPF events not to be limited by ammonia. The
average growth rate for Harwell was found to be 3.37 nm h$^{-1}$, within the range given by Charron et
al. (2007) and higher at N. Kensington at 4.22 nm h$^{-1}$, a trend found for all seasons (Figure 3). The
increased growth rate in the urban area can be related to the greater presence of organic matter and
other condensable species. In both areas NPF events had higher growth rates in summer than in
spring, as was also found in previous studies (Kulmala et al., 2004; Nieminen et al., 2018). This
may be associated with the higher presence of organic compounds emitted by trees during summer
(Riipinen et al., 2007), or faster oxidation rates due to higher concentrations of hydroxyl radical and
ozone (Harrison et al., 2006).

About 45% of the events took place simultaneously in both background areas. These events are
characterized as regional, as NPF takes place in larger scale, regardless of the local conditions on
the given area. In this case, meteorological conditions were even clearer, indicative of the greater
dependence of regional events on synoptic conditions rather than local. While most chemical



constituents were also lower during regional events, different patterns were found for organic
compounds and sulphate for each background area. In Harwell sulphate was higher during regional
events, while in N. Kensington organic compounds were higher during regional events. This may be
indicative of the variable role that specific chemical species have in condensational nanoparticle
growth (Yue et al., 2010). In all cases though, the concentrations of these species were lower
compared to the average conditions. Despite these differences, the growth rate of particles was
found to be higher for local events in N. Kensington (4.4 nm h$^{-1}$) compared to regional events (3.9
nm h$^{-1}$). In Harwell, no difference was found in the growth rate between regional and local events.

**3.1.2    Variability of the origin of the air masses on NPF events**
As both background areas are relatively close to each other and had similar number of event days, a
combined clustering of back trajectories for the event days (only) in these two areas was attempted.
This would provide an insight into the origin of air masses for local and regional events, as well as
the conditions for these air masses. The data for local N. Kensington events and both local and
regional events in Harwell were clustered together and the results along with the characteristics of
the air mass clusters are found in Figure S2.

Cluster C3, which is placed between C2 and C4 among those originating from the Atlantic Ocean,
has the highest percentage for both area specific and regional events. Specifically, for regional
events the percentage is over 35%, much higher compared to all other, showing a clear "preference"



of regional events for cleaner and faster moving air masses from mid-latitudes of the Atlantic
Ocean. This "preference" explains the lower production and growth rate of the new particles found
for regional events, compared to local ones, as air masses from this area have lower organic carbon
and $SO_2$ concentrations. Cluster C5, originating straight from the north but representing air masses
that have crossed the Irish Sea and have not extensively gone over land presents a similar case.
These cold and clean air masses are associated with a low growth rate and survivability of the
newly formed particles. Local events for both sites apart from those in Cluster C3 are highly
associated with Clusters C1 and C2. C1, which contains slow and polluted air masses, presents the
highest growth rate and as a result high particle survivability, as given by the P parameter (see
later). On the other hand, C2 which consists of warm and moist air masses from lower latitudes is
the least common for regional events and presents high growth rate and survival probability of the
particles. Apart from the weak relation found with particulate organic carbon concentrations and
growth rate (Figure S2), there appears to be an inverse relation between the temperature and
survivability of the particles. Warmer air masses seem to be related to higher particle survival
probability, which may be attributable to greater growth rates as temperature increases (Yli-Juuti et
al., 2011).

**3.1.3    Urban increment and particle development**
The urban environment, depending on the conditions, may have a positive or negative effect in the
number of the particles formed and their consequent survival and growth. Both Harwell and N.





Kensington are in background areas, rural and urban respectively. As a result, while the
concentrations of pollutants are higher in N. Kensington than Harwell, their effect is smaller
compared to that of Marylebone Road. A comparison of the particles smaller than 20 nm, gives
insight into the formation and survival of the newly formed particles in the initial stages.
Calculating the urban increment (equation 5) using the two background sites showed around 20%
more particles of size 16 - 20 nm in N. Kensington than Harwell for event days, an increment that is
even stronger when solely local events are considered (Figure 4). As the sizes of the particles in the
calculation are relatively large and due to the higher condensation sink found in N. Kensington, this
increment is expected to be larger for smaller size particles. A possible explanation for this result
may be the greater concentration of organic compounds which is observed in N. Kensington, as
discussed earlier, which leads to more rapid formation of secondary condensable species that
enhances the nucleation process in the more polluted area.

Considering the local events, most of the pollutant data available appear to be higher which is
reflected in the condensation sink as well. The role of the polluted background appears to be
decisive in the further growth of the newly formed particles, especially for Harwell. This, at both
sites causes the number of particles of greater size to be smaller for the later hours in the days of
local events (Figure S3). Another possible reason for this difference in the larger size ranges can be
the higher concentration of organic content on the days of regional events at N. Kensington (as
discussed earlier). On the other hand, for Harwell all hydrocarbons with available data are lower



throughout the day (apart from ethane) during regional events. Unlike N. Kensington, at Harwell
particles smaller than 20 nm as well as the growth rate of the newly formed particles are almost the
same for regional and local events.

The calculation of the increment in Marylebone Road provided negative results; particles smaller
than 20 nm were less abundant on event days compared to the average, throughout the day. This is
due to the fact that Marylebone road is heavily affected by traffic pollution and on average,
conditions do not promote NPF events due to the high condensation sink, unless clear conditions
prevail, which are also associated with a low particle load.

**3.2    NPF Events at Marylebone Road**
For many years, NPF events were thought not to take place in heavily polluted urban areas, as the
effect of the increased condensation sink was considered detrimental in suppressing the formation
and growth of new particles. Recent long term analyses have shown this is not the case and
nowadays an increasing number of studies studies confirm the occurrence of NPF events in urban
areas. In this study, for the same period of seven years as for the two background areas, NPF events
were found to occur for 6.1% of days at Marylebone Road, lower than in the background areas.
Seasonal variation is similar to that at the background sites, but day of the week variation is stronger
at Marylebone Road further favouring weekends (Figure S4), as on these days traffic intensity is
lower.





In general, similar conditions found in the background areas to affect NPF events are also found at
Marylebone Road, despite a much larger condensation sink. (Figure S1). As a result, less particles
of size smaller than 20 nm were found on NPF event days than the average for the site, as the sum
of background particles plus those formed on these days were less than that on an average day. The
growth rate of the newly formed particles is higher than that of the background sites (5.5 nm h$^{-1}$),
which is in agreement with the findings in the study of the background areas on the possible role of
the condensable species, the concentrations of which are even greater at the urban kerbside. At
Marylebone Road, the number of NPF days which were common with the background sites was
fewer, as local conditions (high condensation sink) are detrimental to the occurrence of NPF events
and thus the days of regional events including Marylebone Road were separately studied for this
site. The regional event days that were common for all three sites were 37 (31% of events at
Marylebone Road) (Table 1). As with the other two areas, the growth rate is higher during local
events, but the conditions are mixed, with lower concentrations of sulphate and organic compounds
but higher $SO_2$, NOx and elemental carbon. The relationship with higher wind speed (mainly
western) (Figure S5), solar radiation (which results in greater $H_2SO_4$ formation) and lower relative
humidity, indicate the stronger relation of the regional events with synoptic conditions than the
local events in the heavily polluted environment of Marylebone Road.





### 384 3.3 Connection of NPF Events with Incoming Air Masses

### 385 3.3.1 Air mass back trajectory clustering and connection with NPF events

The origin of the air masses plays a very important role in the occurrence of NPF events, as shown
in Section 3.1.2. Air masses of different origins have different characteristics. Back trajectories
provide excellent insight into the source of the air masses. Air mass back trajectories were
calculated both for all days and for NPF event days for each site separately, with the aim of
complementing the analysis in Section 3.1.2 which addressed only the event days. The additional
analysis gives a view of the frequency of NPF events within different air mass types. The initial air
mass back trajectory clustering ended up with an optimal solution of 9 clusters of different air
masses. As many of these clusters had similar characteristics and origin, solutions with fewer
clusters were attempted. As the number of clusters was decreasing clusters became a mixture of
different origins, thus making the distinction of different sources harder. As a result, the method
chosen was to merge clusters of similar origin and characteristics, which kept the detail of the large
number of clusters and made the separation of the different origins more distinct.

The resulting four merged clusters (Figure 5), using the characterisation proposed by McIntosh et
al., 1969, are:
• An **Arctic** cluster, which originates mainly from the northerly sector. It occurs about 10% of
the time and consists of cold air masses, which either passed over northern parts of the U.K. or
through the Irish Sea.





- A **Tropical** cluster, which originates from the central Atlantic. It occurs 25% of the time and contains warmer air masses. A small percentage of this cluster contains masses that have passed over countries south of the U.K. Even though these days were more polluted, the clustering method was unable to clearly distinguish these days as it does not take into account particle numbers or composition, even when the 9-cluster solution was applied.

- A **Polar** cluster, which originates from the north Atlantic. It is the most common type of air mass arriving in the areas of study and occurs about 40% of the time bringing fast moving, "clean" air masses with increased marine components (Cl, Na, Mg) from the west. This cluster also contains airmasses that have passed through Ireland, though an effect on particle size and chemical composition is not distinct.

- A **Continental** cluster, which originates from the east. It occurs about 25% of the time and consists mainly of slow moving air masses, originating from the London area (for the background areas) and/or continental Europe. It has higher concentrations of most pollutants as well as the highest condensation sink.

The occurrence of each air mass class for average and event days for Harwell and London (both sites) can also be found in Figure 5, while their main characteristics for each site can be found in Table S1. Though in this case the air mass grouping for each site was done in a different analysis, the resulting groups are almost identical in their characteristics and frequency, as the sites are close to each other.



The Polar cluster is the one prevailing on both average and event days. This consists of clean fast-moving air masses originating mainly from mid and high latitudes of the Atlantic, and this cluster presents favourable conditions for NPF events. The association of NPF events with air masses from the mid-Atlantic at N. Kensington was also found by Beddows et al. (2015). Cool Arctic air masses on average are not clean as they may have passed over the northern U.K. The event days associated with this air mass type have the lowest concentrations of the pollutants within available data for all areas. The increased percentage of events with this air mass at all sites indicates that lower temperatures, in a clear atmosphere with sufficient solar radiation are favourable for NPF events as found in previous studies (Napari et al., 2002; Jeong et al., 2010; Kirkby et al., 2011). A similar trend of increased probability with polar and arctic maritime air masses was also found for Hyytiälä, Finland by Nilsson et al. (2001). Tropical air masses have a lower probability for NPF events, which is associated with the fact that a number of these days are associated with air masses which have passed from continental areas south of the U.K. (France, Spain etc.). Specifically for Marylebone Road the NPF probability is a lot lower (11% versus 17% for N. Kensington and 20% for Harwell). This is due to the fact that these air masses are more related to southerly winds which, in Marylebone Road are associated with a street canyon vortex which causes higher pollutant concentrations at this site. Finally, the Continental cluster presents the lowest probability for NPF events. The air masses in this group originate from continental Europe and for the background areas in most cases have passed over the London region as well. This results in both a higher condensation sink and concentration of pollutants, which limits the number of days with favourable



conditions for NPF events. Growth rate for all sites though appears to be higher for air masses
originating from more polluted areas (Figure 6), which appear to enhance the growth process due to
containing a higher concentration of condensable species (after oxidation).

**3.4      Nucleation Strength Factor (NSF)**
The NSF (equations 6 and 7) is used to describe the effect nucleation events have on the number of
particles at a site. The values of NSF for each site and for seasons spring and summer are shown in
Table 2. The decrease of the contribution of NPF events to particle number, moving from the rural
area to the kerbside was also found in previous studies (Salma et al., 2014; 2017). This is explained
by the increased contribution to the particle number concentrations of other sources, mainly
combustion in the urban environment, compared to rural areas. Apart from this trend, in the
background areas the increase of $N_{16-100}$ was greater in spring than summer. This effect seems
stronger in the urban background area compared to the rural, as in that area the variability of $N_{16-100}$
is greater for event days compared to that of the rural area. On the other hand, the contribution of
NPF events in the longer span, as is illustrated by the $NSF_{GEN}$ appears to favour summer for all
areas, showing the increased formation and survivability of particles in this season.

For Marylebone Road the result for the increase of the $N_{16-100}$ is greater in summer than in spring, in
contrast to what was found for the background sites. This is due to the fact that in summer the
traffic intensity is decreased, giving the contribution from NPF events a stronger effect compared to




the other sources. The very small increase found on NPF events in Marylebone Road, with a factor
of just 1.26, a lot lower than that found in the urban area of Seoul, South Korea (Park et al., 2015),
is indicative of the reduced effect of NPF events in an area which is heavily affected by traffic, as
also pointed out by Von Bismarck-Osten et al. (2013) in their study on particle composition in
Marylebone Road.

**3.5      The Survival Parameter P**
The survival parameter P is a measure of the probability for newly formed particles to survive to
detectable sizes. The average values of the P parameter for each of the areas of this study are 10.5
for Harwell, 15.8 for N. Kensington and 28.9 for Marylebone Road. The values found put
Marylebone Road to the upper end of heavily polluted areas in Europe, North Kensington to the
same level as many other urban areas in Europe, while Harwell had somehow higher values
compared to other rural background areas in Europe, as calculated by Kulmala et al. (2017). The
seasonal, air mass origin and local versus regional variations can be found in Figure 7 (winter is
excluded due to very low number of events). While the increasing trend of the P parameter as we
move from rural background to kerbside was expected, it can be seen that there is a clear seasonal
pattern in all three areas, with summer having the lowest P parameter (greatest survivability)
compared to the other two seasons. This is associated with the higher growth rate found in summer
for all areas of this study, as the differences in the condensation sink on event days are negligible
between seasons. The case is similar for regional and local events. The result per air mass origin is



related to the different conditions and parameters of each incoming air mass in each area. For
example, the higher P parameter for Tropical air masses at Marylebone Road, is associated with the
higher condensation sink found for this kind of air masses, due to the street canyon effect which is
specific for Marylebone Road for southerly wind directions with which these air masses are mainly
related, while the higher values for the rather clean Arctic air masses for the other two areas are
associated with the lower growth rates found for this kind of air mass in these areas. The more
polluted Continental air masses seem to have a different effect for rural and urban areas. Their
higher condensation sinks and concentrations of pollutants have a negative effect on P-values for
the rural site and a positive effect at the urban sites. The exact opposite is found for the cleaner air
masses of the Polar cluster, which appear to result in reduced P-values of the newly formed
particles at the urban sites. This is related to the lower condensation sink associated with this air
mass type.

**4.      CONCLUSIONS**
Seven years of data from three distinct areas (regional background, urban background, kerbside) in
the southern U.K. were analysed and the conditions associated with NPF events were studied. NPF
events were found to occur on about 7% of days at background sites and less at the kerbside site.
The conditions on event days for all three areas were similar, with clear atmospheric conditions and
a lower condensation sink. While the condensation sink appears to be the most important factor
limiting NPF events at the kerbside site, $SO_2$ was found to have smaller concentrations on event



days for all areas, which indicates that on average it is in sufficient concentrations for NPF events to
occur. The growth rate of the newly formed particles increases from the rural site to the kerbside
and is greater in summer compared to other seasons for all three sites. Almost half of the NPF
events at the rural and urban background sites were found to happen simultaneously. In these cases,
the atmospheric conditions were cleaner, which resulted in slower growth rates. While most of the
chemical species available were at lower concentrations in regional events, a difference in the
behaviour with respect to sulphate and organic compounds was found between the two background
site types.

The prevailing origin of air masses in the southern U.K. is from mid and high latitudes of the
Atlantic Ocean. These fast-moving air masses present an increased probability for NPF to occur.
The case is similar for the cooler and cleaner arctic air masses, while air masses from the tropics
and continental Europe, having greater pollutant content, have decreased NPF probability, but a
higher growth rate of particles when NPF events occurred. Regional events appear to be more
associated with cleaner air masses, presenting a smaller growth rate and condensation sink
compared to local events. The difference in growth rate is probably related to the greater content of
condensable species; a positive relation of particle survival probability with temperature was also
found.




Comparing the background areas in this study, particles of 16-20 nm were found to be about 20%
greater in concentration (above long-term average) on NPF event days at the urban backbround site
compared with the rural site. This is associated with a higher abundance of condensable species in
the urban environment, which enhances the nucleation and growth process. This effect though is
limited as particle size increases and NPF events have a greater effect on the overall $N_{<100\,nm}$ in the
rural areas, compared to urban, as calculated by the NSF. The effect becomes even smaller at the
kerbside as the number of background particles emitted by traffic is a lot greater.

The occurrence of NPF events at the highly polluted Marylebone Road site is at first sight
surprising given the elevated condensation sink.  This must be counteracted by an abundance of
condensable material, which is surprising given the generally modest rate of atmospheric oxidation
processes in comparison to residence times in a street canyon (Harrison, 2017).  However, Giorio et
al. (2015), using Aerosol Time-of-Flight Mass Spectrometry, reported rapid chemical processes
within the Marylebone Road street canyon leading to production of secondary particulate matter
from road traffic emissions.  They postulated that this resulted from very local gas to particle
conversion from vehicle-emitted pollutants.  Condensation of such reaction products upon pre-
existing particles could explain the enhanced particle growth rates observed at Marylebone Road
(Figure 3).



Finally, particle survival probability was found to decrease moving from rural to urban areas. While
formation and initial growth of new particles is increased in urban areas, their survivability reduces
as their size increases. The probability of particles to survive to greater sizes was found to be
increased in summer for all areas, which is also explained by the higher growth rate. The probability
is also different depending upon the origin of the air masses and is related to conditions specific for
each area.

In the present work, the effects of atmospheric conditions upon the NPF process are studied. NPF is
a complex process, highly affected by meteorological conditions (local and synoptic), the chemical
composition as well as the pre-existing conditions in an area. For this reason, the study of NPF
events in one area cannot provide safe assumptions for other areas, as the mixture of conditions
found in different places is unique and alters the occurrence and development of NPF events. Thus,
more studies on the conditions and the trends in NPF events should be conducted to better
understand the effect of the numerous variables that affect those processes.


**AUTHOR CONTRIBUTIONS**
This study was conceived by MD and RMH who also contributed to the final manuscript.  The data
analysis was carried out by DB with guidance from DCSB, and DB also prepared the first draft of
the manuscript. FDP provided advice on the analysis.





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





**TABLE LEGENDS:**

**Table 1:**     Number of NPF events per site.

**Table 2:**     Annual and seasonal NSF for all areas of study.

**FIGURE LEGENDS:**

**Figure 1:**     Map of the measuring stations.

**Figure 2:**     Number of NPF events per season (Winter – DJF;  Spring – MAM;  Summer – JJA;
Autumn – SON) at Harwell (rural), N. Kensington (urban background) and
Marylebone Road (urban roadside).

**Figure 3:**     Growth rate per season at the three sites.

**Figure 4:**     Diurnal variation of $N_{16-20nm}$ at each site:  annual average and NPF event days.

**Figure 5:**     Map and frequency of incoming air mass origin – average and for NPF events per site.

**Figure 6:**     Growth rate per incoming air mass at each of the sites.

**Figure 7:**     Survival parameter P (a) per season, (b) for regional and local events (for Marylebone
Road) is regional for all 3 sites and (c) by incoming air mass origin.



**Table 1:** Number of NPF events per site.

|  | **Harwell** | **N. Kensington** | **Marylebone Road** | **Regional (Background sites)*** | **Regional (All 3 sites)**** |
|---|---|---|---|---|---|
| 2009 | 9 | 0 | 4 | 0 | 0 |
| 2010 | 29 | 22 | 22 | 11 | 9 |
| 2011 | 15 | 10 | 23 | 4 | 1 |
| 2012 | 8 | 28 | 12 | 3 | 0 |
| 2013 | 25 | 23 | 27 | 13 | 11 |
| 2014 | 29 | 34 | 13 | 18 | 6 |
| 2015 | 25 | 22 | 18 | 11 | 10 |
| Overall | 140 | 139 | 119 | 60 | 37 |

* Refers to events occurring simultaneously at Harwell and N. Kensington
** Refers to events which occur simultaneously at all three sites



**Table 2:** Annual and seasonal NSF for all areas of study.

|  | **Harwell** | **N. Kensington** | **Marylebone Road** |
|---|---|---|---|
| $NSF_{NUC}$ (Spring) | 2.04 | 2.03 | 1.2 |
| $NSF_{NUC}$ (Summer) | 2.01 | 1.72 | 1.26 |
| $NSF_{NUC}$ (Year) | 2.25 | 1.86 | 1.26 |
| $NSF_{GEN}$ (Spring) | 1.1 | 1.07 | 1.02 |
| $NSF_{GEN}$ (Summer) | 1.18 | 1.11 | 1.01 |
| $NSF_{GEN}$ (Year) | 1.1 | 1.06 | 1.02 |






**Figure 1:** Map of the measuring stations.



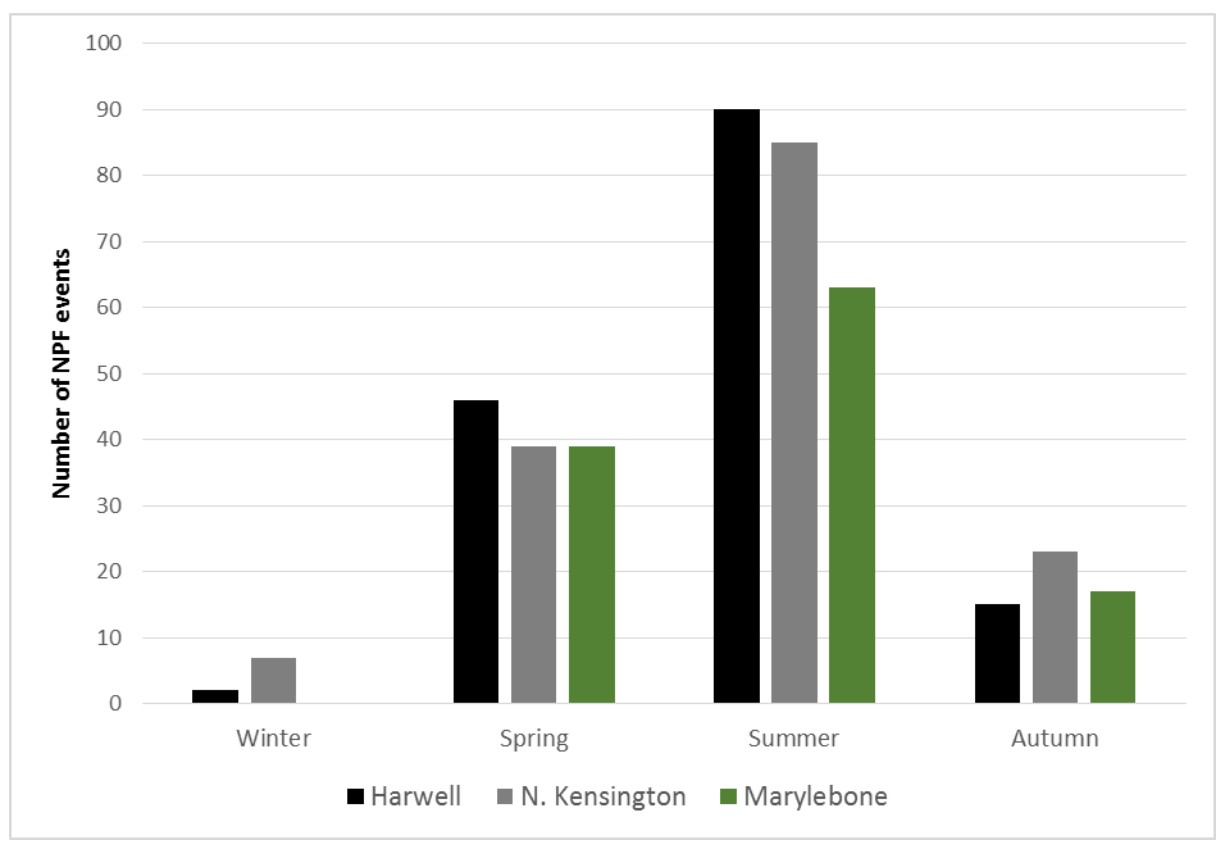


**Figure 2:** Number of NPF events per season (Winter – DJF;  Spring – MAM;  Summer – JJA; Autumn – SON) at Harwell (rural), N.Kensington (urban background) and Marylebone Road (urban roadside).




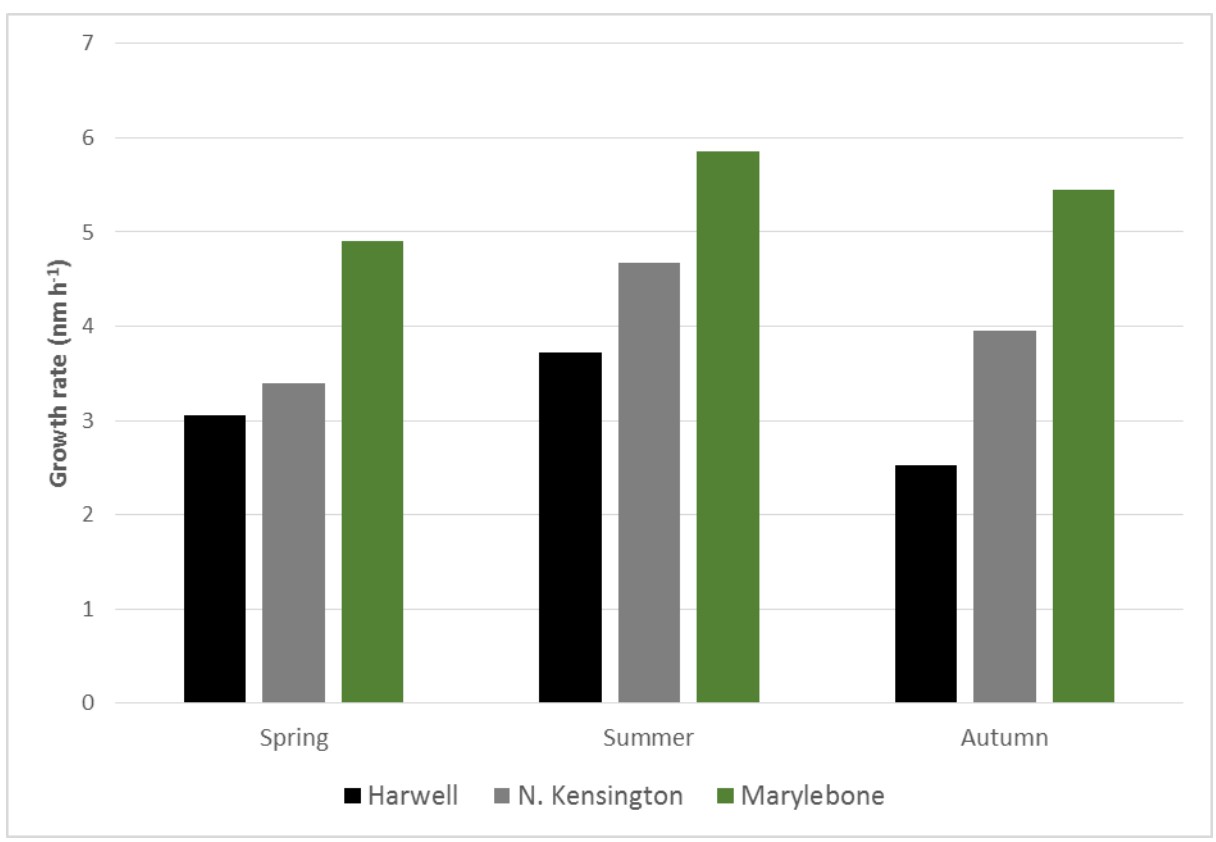

**Figure 3:** Growth rate per season at the three sites.




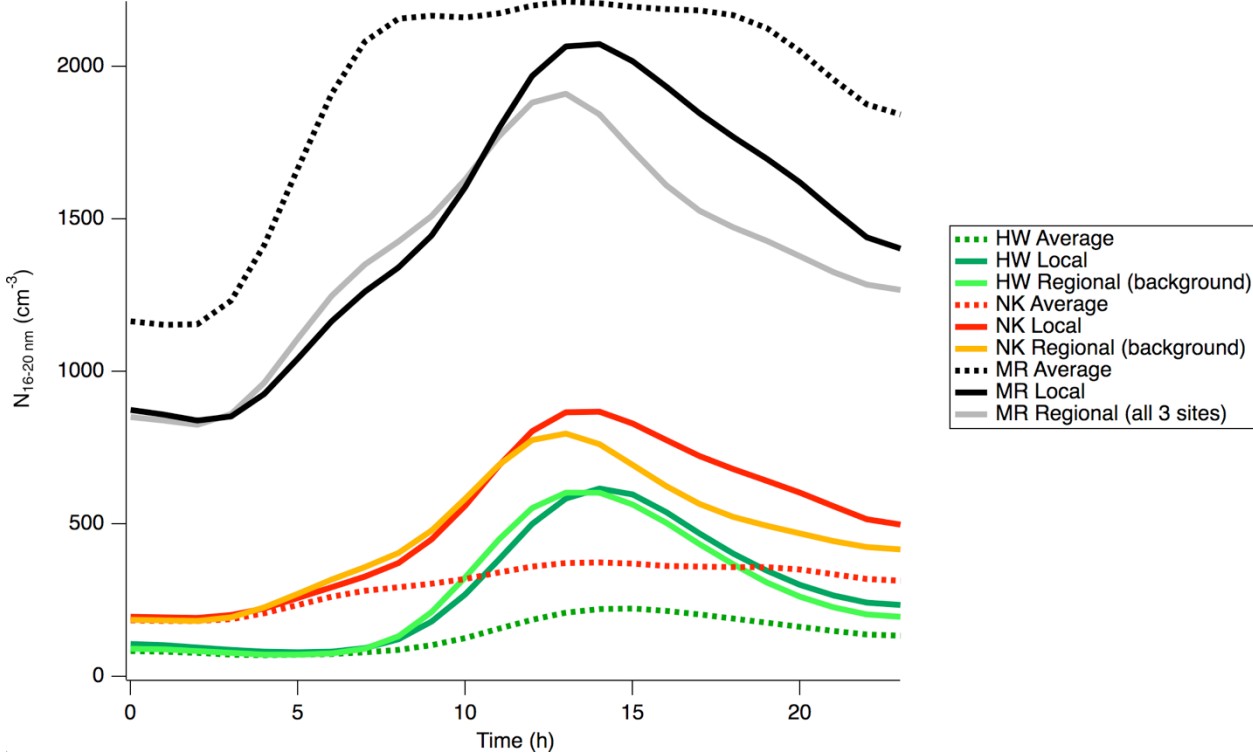

**Figure 4:** Diurnal variation of $N_{16-20nm}$ at each site: annual average and NPF event days.








**Figure 5:** Map and frequency of incoming air mass origin – average and for NPF events per site.






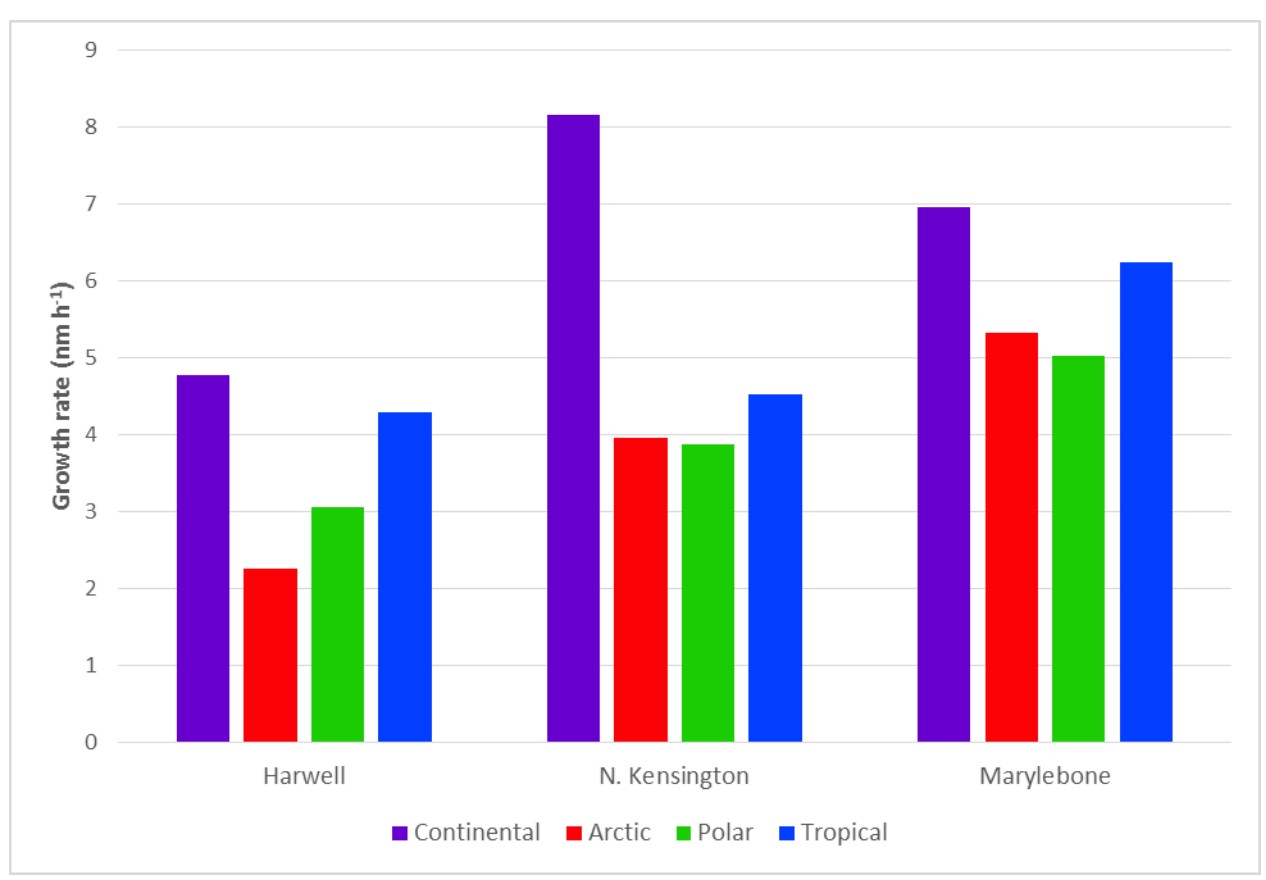

**Figure 6:** Growth rate per incoming air mass origin at each of the sites.





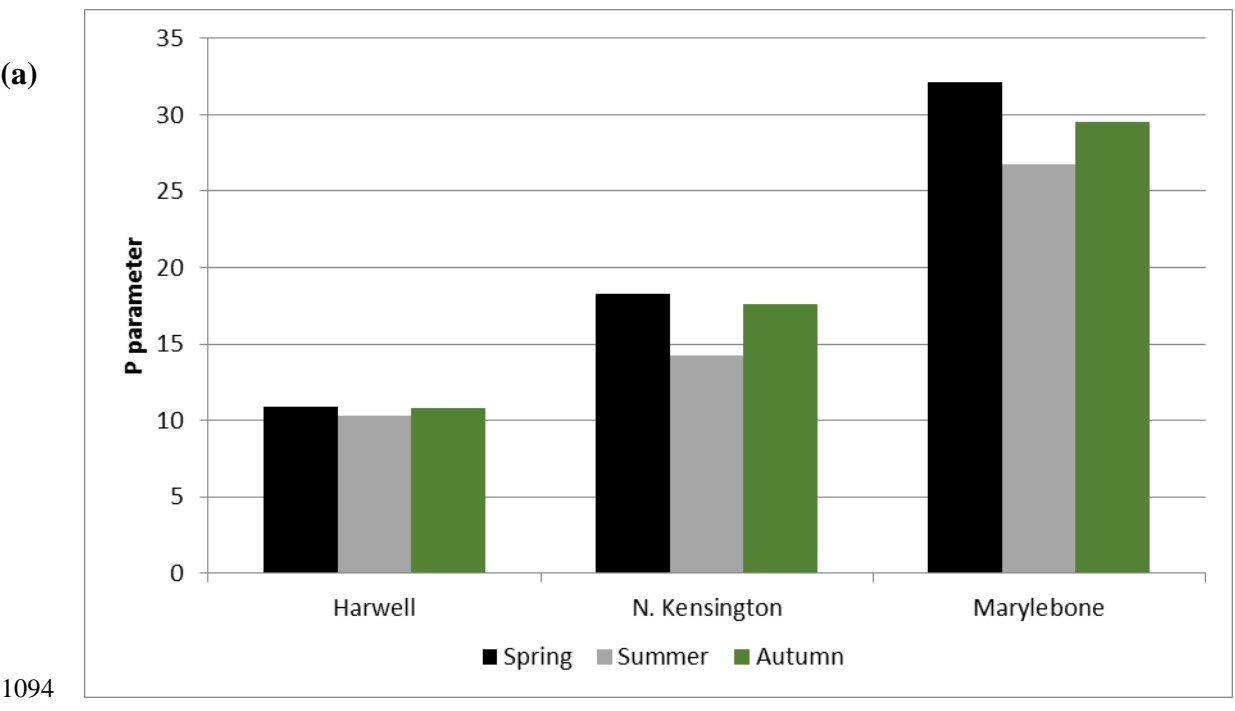










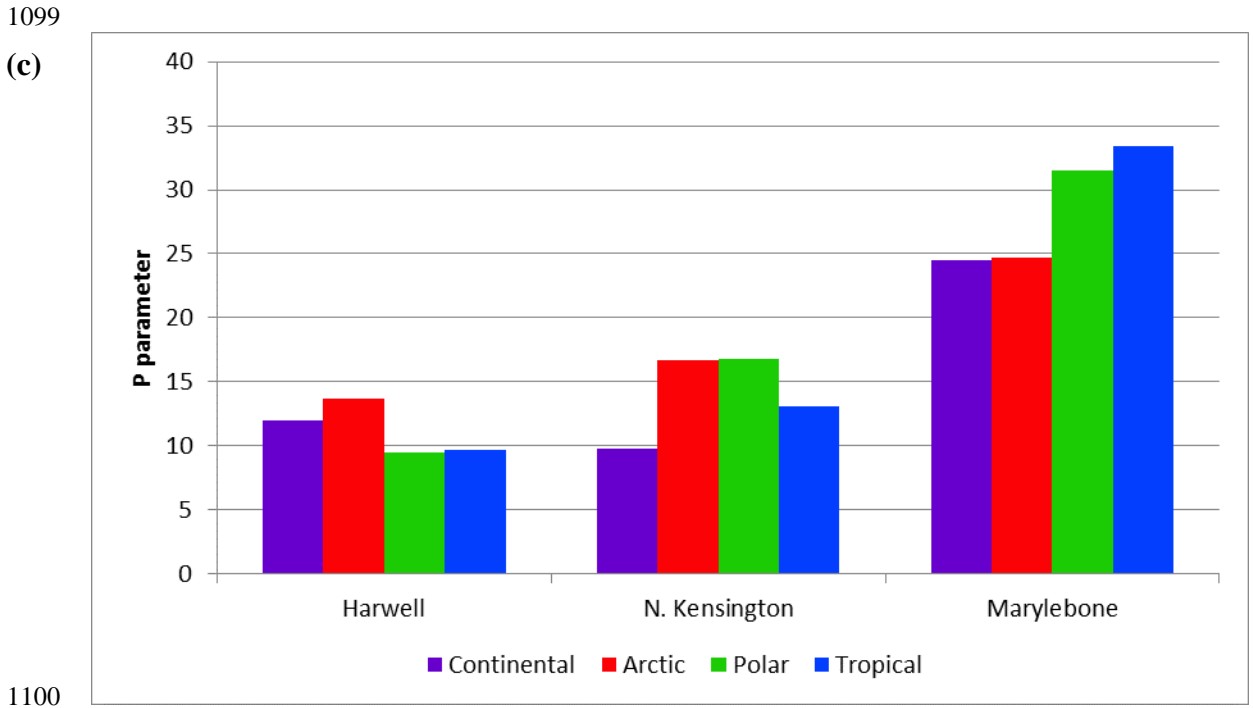


**Figure 7:** Survival parameter P (a) per season, (b) for regional and local events (for Marylebone
Road regional is for all 3 sites) and (c) by incoming air mass origin.
