# Peer review of "Analysis of New Particle Formation (NPF) Events at Nearby Rural, Urban Background and Urban Roadside Sites"

_Atmospheric Chemistry and Physics, 2018_

## Referee Comment (RC1) · Anonymous Referee #1 · 20 Dec 2018

The MS mainly deals with the occurrence frequency, particle growth rate, condensation sink, nucleation strength factor, survival parameter and relationships among them at 3 different locations (rural, urban background and urban roadside sites) in the UK over several years. It contains valuable results and conclusions. Some parts of the MS should be elaborated better (some items are given below as examples), and they can definitely be handled and improved. There is, however, a conceptual weakness of the study related to the lower diameter limit of the SMPS system (of 16.6 nm) which can represent the largest source of inconclusive or ambiguous interpretations for the urban sites.

**Major comment**

1. New particle formation and growth events are mainly identified, separated from emission sources and classified on the basis of particle number size distributions in the particle diameter range <20 nm (e.g. Kulmala et al., Nat. Protoc., 7, 1651–1667, 2012). The diameter interval available for this in the evaluated work, namely 16.6–20 nm is quite narrow in particular, when you consider the logarithmic scale of the abscissa of size distributions. More importantly, the lower limit is requested to be even smaller (preferably below 10 nm or at 3 nm) for studies in urban atmospheric environments, where huge emission peaks can temporary dominate the smallest size ranges as well (Nieminen et al., Atmos. Chem. Phys., 18, 14737–14756, 2018). This property (16.6 nm lower limit) of the measuring system and its consequences for the data treatment, results and conclusions at the urban sites should definitely be discussed in detail, explained and resolved before any further opinion could be formed or decision can be made.

**Some minor comments**

1. Lines 21, 69, etc.: it is advised not to start a sentence with abbreviation.
2. Line 61: consider writing primary particles or emission sources instead of primary emissions.
3. Lines 106–109: it is unusual to attribute particles with a diameter between 1.3 and 3 nm to road traffic emissions, and, therefore, this should be discussed and explained in more detail.
4. Lines 149–151 or Table 1: supply more detailed data coverage, e.g. for each year or season of years.
5. Lines 198–203: it is requested that the diameter of particles under consideration is specified as the growth rate changes with diameter.

6. Lines 262, 263, Table 2: revisit your rounding off strategy.

7. Lines 462–463: remove; it is a repetition from lines 235–239.

8. Fig. 2: it is unclear from the figure or related text which time interval was considered here. A number of NPF events of 90 at Harwell in summer (JJA, 92 days) should be clarified to avoid any misunderstanding.

---

## Referee Comment (RC2) · Anonymous Referee #2 · 22 Dec 2018

Manuscript entitled 'Analysis of New Particle Formation (NPF) Events at Nearby Rural, Urban Background and Urban Roadside Sites' by Bousiotis et al. reports the occurrence of new particle formation events at three sites of different environments in the United Kingdom: Rural, urban background and near road sites. The authors study parameters of new particle formation such as frequency, growth rates, number concentration of sizes 16 – 20 nm, condensation sink, urban increment, nucleation strength factor and survival probability. The authors also report trajectory cluster analysis as well as the connection of NPF between the three different sites. In general, the manuscript, contains valuable data (three sites of different environments) and treasured statistics (7 years of data). In addition, the manuscript is well written and literature from around the world is acknowledged. However, the authors make big assumptions and conclusions without enough supporting data. The major concerns listed below need to be addressed before the manuscript is considered for publication in ACP.

Major Comments:

1. The authors report the observation of new particle formation events at three sites in the UK based on visual inspection of CPC (> 7 nm) and SMPS (> 16.6 nm). The general character of NPF events is missing. The lowest limit of the instrument is an issue and no big conclusions can be made before ensuring that the observed plume of particles is related to a new particle formation event. Authors should report how these events look like and whether they have a growing mode shape. Also, more characteristics of the growth should be reported such as possible shrinkage (see e.g. Salma et al. (2016)) and the size these particles reach. An example surface plot from each site should be added to the manuscript.

   Let's take for example a regional event surface plot from Kerminen et al. (2018): figure 1, if we cannot observe the information below 16 nm, how can the authors prove that the increase in particle concentration is related to NPF, figure 2.

[Figure]

*Figure 1 Regional Event example. Figure from Kerminen et al 2018.*

[Figure]

*Figure 2 Modified figure 1.*

The manuscript refers to many NPF studies from around the world, many of which report NPF starting from 6 nm (Salma et al., 2017), 3 nm (Dal Maso et al., 2005), 1.7 nm (Kirkby et al., 2016) while their measurement starts from 16.6 nm. The authors should present evidence that these observed particles are related to new particle formation events, and not for example a traffic growing mode (Brines et al., 2015).

2. Section 2.1: Which years are studied?

3. Section 2.1: Distance between the three sites should be mentioned.

4. Section 2.2.1: Authors report a visual inspection of CPC and SMPS data.
   - How was this exactly done? Please elaborate.
   - Was there any kind of counter-calibration done between these instruments?
5. Section 2.2.2: Calculation of the growth rates:
   - Size of growth rates should be mentioned. E.g. growth from 7 to 20 nm? To 50 nm?
   - How many points were taken in calculating the GR?
   - Line 290: Authors claim that GR in NK (4.4 nm/h) are higher than the regional events GR (3.9 nm/h), what is the error bar on these calculations? Accordingly, these growth rates might be similar.
6. On line 178: the author mention nucleation mode, which is by definition number of particles between 3 and 25 nm, while the authors conduct a large study on a small fraction of this nucleation mode ( $16 - 25$ nm).
7. Section 2.2.4: Reference to Kulmala et al. 2017, calculating P = CS'/GR. What GR was used here? See point 4.
8. Section 3.1.1: Reference to Figure S1: cloudiness, and RH…. Is missing. Was cloudiness measured or calculated?
9. Section 3.1.2: How can the authors prove that NPF events are happening at the near road site and not transported to the location?
10. The authors make big conclusions regarding the $SO_2$ driving mechanism of NPF which cannot be proved without adequate chemical speciation of the particles formed. These conclusions shall be minimized throughout the manuscript. Authors could try calculating sulfuric acid proxy from SO2 and CS (Petäjä et al., 2009).

**References**

Brines, M., Dall'Osto, M., Beddows, D., Harrison, R., Gómez-Moreno, F., Núñez, L., Artíñano, B., Costabile, F., Gobbi, G., Salimi, F. J. A. C., and Physics: Traffic and nucleation events as main sources of ultrafine particles in high-insolation developed world cities, 15, 5929-5945, 2015.

Dal Maso, M., Kulmala, M., Riipinen, I., Wagner, R., Hussein, T., Aalto, P. P., and Lehtinen, K. E.: Formation and growth of fresh atmospheric aerosols: eight years of aerosol size distribution data from SMEAR II, Hyytiala, Finland, Boreal Env. Res., 10, 323, 2005.

Kerminen, V.-M., Chen, X., Vakkari, V., Petäjä, T., Kulmala, M., and Bianchi, F. J. E. R. L.: Atmospheric new particle formation and growth: review of field observations, Environmental Research Letters, 13, 103003, ARTN 10300310.1088/1748-9326/aadf3c, 2018.

Kirkby, J., Duplissy, J., Sengupta, K., Frege, C., Gordon, H., Williamson, C., Heinritzi, M., Simon, M., Yan, C., and Almeida, J. J. N.: Ion-induced nucleation of pure biogenic particles, Nature, 533, 521, 10.1038/nature17953, 2016.

Petäjä, T., Mauldin Iii, R., Kosciuch, E., McGrath, J., Nieminen, T., Paasonen, P., Boy, M., Adamov, A., Kotiaho, T., and Kulmala, M.: Sulfuric acid and OH concentrations in a boreal forest site, Atmos. Chem. Phys., 9, 7435-7448, 10.5194/acp-9-7435-2009, 2009.

Salma, I., Németh, Z., Weidinger, T., Kovács, B., Kristóf, G. J. A. C., and Physics: Measurement, growth types and shrinkage of newly formed aerosol particles at an urban research platform, Atmospheric Chemistry and Physics, 16, 7837-7851, 10.5194/acp-16-7837-2016, 2016.

Salma, I., Varga, V., Németh, Z. J. A. C., and Physics: Quantification of an atmospheric nucleation and growth process as a single source of aerosol particles in a city, 17, 15007-15017, 2017.

---

## Author Response (AR1)

Journal: ACP - MS No.: acp-2018-1057
Title: Analysis of New Particle Formation (NPF) Events at Nearby Rural, Urban Background and Urban Roadside Sites
Author(s): Dimitrios Bousiotis et al.

**RESPONSE TO REVIEWERS**

We thank the reviewers for their valuable comments, and respond point by point below.

**ANONYMOUS REFEREE #2**

Manuscript entitled 'Analysis of New Particle Formation (NPF) Events at Nearby Rural, Urban Background and Urban Roadside Sites' by Bousiotis et al. reports the occurrence of new particle formation events at three sites of different environments in the United Kingdom: Rural, urban background and near road sites. The authors study parameters of new particle formation such as frequency, growth rates, number concentration of sizes 16 – 20 nm, condensation sink, urban increment, nucleation strength factor and survival probability. The authors also report trajectory cluster analysis as well as the connection of NPF between the three different sites. In general, the manuscript, contains valuable data (three sites of different environments) and treasured statistics (7 years of data). In addition, the manuscript is well written and literature from around the world is acknowledged. However, the authors make big assumptions and conclusions without enough supporting data. The major concerns listed below need to be addressed before the manuscript is considered for publication in ACP.

**Major Comments:**

1. The authors report the observation of new particle formation events at three sites in the UK based on visual inspection of CPC (> 7 nm) and SMPS (> 16.6 nm). The general character of NPF events is missing. The lowest limit of the instrument is an issue and no big conclusions can be made before ensuring that the observed plume of particles is related to a new particle formation event. Authors should report how these events look like and whether they have a growing mode shape. Also, more characteristics of the growth should be reported such as possible shrinkage (see e.g. Salma et al. (2016)) and the size these particles reach. An example surface plot from each site should be added to the manuscript.

Let's take for example a regional event surface plot from Kerminen et al. (2018): figure 1, if we cannot observe the information below 16 nm, how can the authors prove that the increase in particle concentration is related to NPF, figure 2.

[Figure]

*Figure 1 Regional Event example. Figure from Kerminen et al 2018.*

[Figure]

*Figure 2 Modified figure 1.*

The manuscript refers to many NPF studies from around the world, many of which report NPF starting from 6 nm (Salma et al., 2017), 3 nm (Dal Maso et al., 2005), 1.7 nm (Kirkby et al., 2016) while their measurement starts from 16.6 nm. The authors should present evidence that these observed particles are related to new particle formation events, and not for example a traffic growing mode (Brines et al., 2015).

**RESPONSE:** The dataset available, as mentioned in the text ranges from 16.6 to 604 nm. To overcome this limitation additional data was used to ensure the correct identification of the NPF events.

To achieve this:

- CPC data was used to provide insight into whether there was an increase on the number of particles of smaller size. An increased number of particles in the size range 7 – 16 nm (provided by the CPC data) right before or at the same time when observed in the SMPS data was a necessary criterion for the occurrence of an event.
- High resolution pollution data was used alongside particle number concentration data in a side by side comparison. A sharp increase in the particle number concentration which was accompanied by a similar increase in the concentrations of pollutants was an indication that these particles were probably associated with pollutant emissions. This was mainly an issue in the roadside (MR) and to a smaller extent with the background sites. Increased particle number concentrations observed at times matching the morning or evening traffic rush hours were also ignored at MR as they always coincided with increased concentrations of pollutants.
- Meteorological data was used. This mainly applies to the urban background site (NK), being in close proximity to London city centre. The possibility of a plume of pollution originating from the London city centre was considered when the site was downwind of it. A power plant to the northeast of the rural background site (HW) was also considered as a possible source of particles, though the distance is larger. Finally, as mentioned in the text, Heathrow airport and its influence were also considered.

In addition to this, the criteria set by Dal Maso et. al. (2005) were fully considered and unless there was a clear new mode of particles at the lower size range of the nucleation mode with a clear growth for at least 3 hours, an NPF event was not assigned. An example of the appearance of the events for each site has been added in the manuscript. Additionally, a discussion of particle shrinkage at later stages, which was observed at MR, is also added to the text.

Due to the limitations of the dataset, events in which the newly formed particles failed to grow to greater than 16 nm could not be seen except in the CPC data. These were rare and due to lack of additional information about their development were ignored. This clarification has been added in the text.

2. Section 2.1: Which years are studied?
**RESPONSE:** The years studied are 2009 – 2015. This information has been added in the text.

3. Section 2.1: Distance between the three sites should be mentioned.
**RESPONSE:** The distance between MR and NK is 4.5 km. The distance between HW and London city centre is about 80 km. This information has been added to the text.

4. Section 2.2.1: Authors report a visual inspection of CPC and SMPS data.
- How was this exactly done? Please elaborate.
- Was there any kind of counter-calibration done between these instruments?
**RESPONSE:** The method used was visual inspection of SMPS data supplemented by the use of CPC data to confirm the increase of the particle number concentration in the smaller size range (7 – 16 nm), as mentioned in (1). The text has been updated to clarify the method used. Both instruments are calibrated by the National Physical Laboratory according to the latest internationally recommended protocols.

5. Section 2.2.2: Calculation of the growth rates:
- Size of growth rates should be mentioned. E.g. growth from 7 to 20 nm? To 50 nm?

- How many points were taken in calculating the GR?
- Line 290: Authors claim that GR in NK (4.4 nm/h) are higher than the regional events GR (3.9 nm/h), what is the error bar on these calculations? Accordingly, these growth rates might be similar.

**RESPONSE:** As the lower size available was 16 nm, a calculation of the growth rate up to 50 nm was chosen (rather than up to 30 nm, which provided poor results in many cases due to the small range). The number of points taken depended on the development of the event and were considered from the start of the event until a) growth stopped, b) GMD reached 50 nm or c) the day ended. These points were added to the manuscript to clarify the method used.

On the third point made, due to the large variation of the growth rates of the events, the error bars are overlapped for the two groups of events. This has been included as a note in the text.

6. On line 178: the author mention nucleation mode, which is by definition number of particles between 3 and 25 nm, while the authors conduct a large study on a small fraction of this nucleation mode ( 16 – 25 nm).

**RESPONSE:** We are not aware of a widely recognised definition of the nucleation mode, with the term taking in different size ranges in the literature. Regardless of that, in the text it is mentioned that *"NPF events are considered when a distinctly new mode of particles which appears in the size distribution at nucleation mode size, prevails for some hours and shows signs of growth",* which is accurate in relation to the criteria set for NPF event selection in this study.

7. Section 2.2.4: Reference to Kulmala et al. 2017, calculating $P = CS'/GR$. What GR was used here? See point 4.

**RESPONSE:** The growth rate and condensation sink used are the ones calculated by the methods mentioned in the text. A clarification of this has been added in the text.

8. Section 3.1.1: Reference to Figure S1: cloudiness, and RH.... Is missing. Was cloudiness measured or calculated?

**RESPONSE:** Cloud amount data, as for all other meteorological data were measurements provided by the Met Office, as mentioned in the text. A plot with average cloud amount for each site has also been added in the supplementary.

9. Section 3.1.2: How can the authors prove that NPF events are happening at the near road site and not transported to the location?

**RESPONSE:** It cannot be stated with certainty whether the NPF took place at the site or particles were advected. What can be said though with confidence is that regardless of where the particle formation took place (either on the spot or in the close vicinity, as particles of that size range cannot travel to distances greater than some kilometers before either reaching detectable sizes or being diluted – especially in a polluted environment such as the London city centre), the new mode not only persists but it also grows for at least 3 hours. A clarification of this has been added in the text. If the events at the roadside site were due to advection, or a purely regional phenomenon, a much closer correlation of event days and growth rates between MR and NK would be expected than was observed.

10. The authors make big conclusions regarding the $SO_2$ driving mechanism of NPF which cannot be proved without adequate chemical speciation of the particles formed. These conclusions shall be minimized throughout the manuscript. Authors could try calculating sulfuric acid proxy from SO2 and CS (Petäjä et al., 2009).

**RESPONSE:** In the text is stated that $SO_2$ was found to be lower on event days compared to the average, which logically leads to the conclusion that either the greater concentrations of $SO_2$ are associated with a more polluted environment with an increased condensation sink (which consequently has a negative effect in the occurrence of an event), or its concentration is adequate and it is not a factor affecting the occurrence of an event (positively or negatively). The calculation of the $H_2SO_4$ proxy was carried out and provided information that did not help in clarifying this point. It was found that the proxy was higher on event days at the background sites and gave an unclear result for the roadside. This result though provides no additional information as the increased values of the proxy are the result of the higher solar radiation and the lower condensation sink found during events. Changes were made in the text to "soften" these conclusions.

**ANONYMOUS REFEREE #1**
The MS mainly deals with the occurrence frequency, particle growth rate, condensation sink, nucleation strength factor, survival parameter and relationships among them at 3 different locations (rural, urban background and urban roadside sites) in the UK over several years. It contains valuable results and conclusions. Some parts of the MS should be elaborated better (some items are given below as examples), and they can definitely be handled and improved. There is, however, a conceptual weakness of the study related to the lower diameter limit of the SMPS system (of 16.6 nm) which can represent the largest source of inconclusive or ambiguous interpretations for the urban sites.

**Major comment:**
1. New particle formation and growth events are mainly identified, separated from emission sources and classified on the basis of particle number size distributions in the particle diameter range <20 nm (e.g. Kulmala et al., Nat. Protoc., 7, 1651–1667, 2012). The diameter interval available for this in the evaluated work, namely 16.6–20 nm is quite narrow in particular, when you consider the logarithmic scale of the abscissa of size distributions. More importantly, the lower limit is requested to be even smaller (preferably below 10 nm or at 3 nm) for studies in urban atmospheric environments, where huge emission peaks can temporary dominate the smallest size ranges as well (Nieminen et al., Atmos. Chem. Phys., 18, 14737–14756, 2018). This property (16.6 nm lower limit) of the measuring system and its consequences for the data treatment, results and conclusions at the urban sites should definitely be discussed in detail, explained and resolved before any further opinion could be formed or decision can be made.
**RESPONSE:** The limitations and consequences due to the available dataset, as well as the additions in the method to ensure the correct selection of NPF events are explained at length in the response to Referee #2 (earlier in this document). As a result of this, clarification of the method and the additional data used have been added to the text.

**Some minor comments:**
1. Lines 21, 69, etc.: it is advised not to start a sentence with abbreviation.
**RESPONSE:** Text updated to address the comment.

2. Line 61: consider writing primary particles or emission sources instead of primary emissions.
**RESPONSE:** Text updated to address the comment.

3. Lines 106–109: it is unusual to attribute particles with a diameter between 1.3 and 3 nm to road traffic emissions, and, therefore, this should be discussed and explained in more detail.
**RESPONSE:** Text updated to accurately reflect the conclusions of the study mentioned.

4. Lines 149–151 or Table 1: supply more detailed data coverage, e.g. for each year or season of years.
**RESPONSE:** Table has been added in the S.I. for detailed seasonal data coverage.

5. Lines 198–203: it is requested that the diameter of particles under consideration is specified as the growth rate changes with diameter.
**RESPONSE:** Text updated to include the size range of the particles considered in the calculation of the growth rate.

6. Lines 262, 263, Table 2: revisit your rounding off strategy.
**RESPONSE:** Text and tables updated to follow a uniform rounding scheme.

7. Lines 462–463: remove; it is a repetition from lines 235–239.
**RESPONSE:** Text updated to remove repeated information.

8. Fig. 2: it is unclear from the figure or related text which time interval was considered here. A number of NPF events of 90 at Harwell in summer (JJA, 92 days) should be clarified to avoid any misunderstanding.

**RESPONSE:** The figure's description has updated to clarify the period plotted.

[revised manuscript text omitted]

Commented [DB(wISAS+19]: Addressing comment 6 (1st)

Commented [DB(wISAS+20]: Addressing comment 6 (1st)

[Figure]

Figure 1: Map of the measuring stations.

[Figure]

**Figure 2:** Number of NPF events per season for all seven years of the present study (Winter – DJF;
Spring – MAM; Summer – JJA; Autumn – SON) at Harwell (rural), N.Kensington (urban
background) and Marylebone Road (urban roadside).

**Commented [DB(wISAS+21]:** Addressing comment 8 (1st)

[Figure]

**Figure 3:** Growth rate per season at the three sites.

[Figure]

**Figure 4:** Diurnal variation of $N_{16-20nm}$ at each site: annual average and NPF event days.

[Figure]

**Figure 5:** Map and frequency of incoming air mass origin – average and for NPF events per site.

[Figure]

**Figure 6:** Growth rate per incoming air mass origin at each of the sites.

[Figure]

[Figure]

(c)

[Figure]

**Figure 7:** Survival parameter P (a) per season, (b) for regional and local events (for Marylebone Road regional is for all 3 sites) and (c) by incoming air mass origin.

**SUPPLEMENTARY INFORMATION**

**Analysis of New Particle Formation (NPF) Events at Nearby Rural, Urban Background and Urban Roadside Sites**

**Dimitrios Bousiotis, Manuel Dall'Osto, David C.S. Beddows and**

**Roy M. Harrison**

**Table S1: Data availability per season (all numbers are percentages of available data)**

| | Harwell | | | | N. Kensington | | | | Marylebone Road | | | |
|---|---|---|---|---|---|---|---|---|---|---|---|---|
| | Winter | Spring | Summer | Autumn | Winter | Spring | Summer | Autumn | Winter | Spring | Summer | Autumn |
| 2009 | 15 | 97 | 10 | 80 | 57 | 97 | 100 | 100 | 100 | 65 | 86 | 68 |
| 2010 | 37 | 53 | 100 | 95 | 58 | 87 | 93 | 100 | 46 | 100 | 87 | 86 |
| 2011 | 72 | 75 | 99 | 73 | 89 | 87 | 73 | 89 | 79 | 99 | 100 | 67 |
| 2012 | 82 | 86 | 100 | 95 | 56 | 88 | 99 | 86 | 0 | 0 | 87 | 66 |
| 2013 | 91 | 70 | 99 | 100 | 84 | 92 | 98 | 98 | 57 | 92 | 84 | 100 |
| 2014 | 97 | 62 | 99 | 99 | 84 | 78 | 97 | 98 | 89 | 79 | 76 | 99 |
| 2015 | 77 | 100 | 61 | 70 | 80 | 99 | 65 | 100 | 74 | 100 | 98 | 100 |

Commented [DB(wISAS+1]: Addressing comment 4 (1st)

**Table S2:** Conditions per air mass origin for NPF event days (April – October average in
parenthesis) for all areas of study.

| Harwell | | | | |
|---|---|---|---|---|
| | Continental | Arctic | Polar | Tropical |
| Condensation sink ($s^{-1}$) | 5.05E-03 (5E-03) | 2.71E-03 (3.32E-03) | 2.57E-03 (2.87E-03) | 3.19E-03 (2.87E-03) |
| Wind speed (m $s^{-1}$) | 3.52 (3.63) | 3.87 (3.47) | 3.64 (3.69) | 3.74 (4.17) |
| Temperature (°C) | 15.5 (13.6) | 12.2 (11.5) | 13.6 (13.1) | 16.3 (15) |
| $SO_2$ ($\mu g\ m^{-3}$) | 1.87 (1.81) | 1.11 (1.82) | 1.11 (1.27) | 1.22 (1.36) |
| $NO_x$ ($\mu g\ m^{-3}$) | 9.58 (13.9) | 5.49 (8.01) | 4.66 (7.2) | 5.81 (7.69) |
| $SO_4^{2-}$ ($\mu g\ m^{-3}$) | 2.70 (3.3) | 1.37 (2.05) | 1.44 (1.64) | 1.37 (1.57) |
| Particulate OC ($\mu g\ m^{-3}$) | 2.85 (2.88) | 1.35 (1.59) | 1.52 (1.63) | 1.98 (1.76) |

| North Kensington | | | | |
|---|---|---|---|---|
| | Continental | Arctic | Polar | Tropical |
| Condensation sink ($s^{-1}$) | 7.20E-03 (9.35E-03) | 5.20E-03 (6.37E-03) | 5.40E-03 (6.38E-03) | 4.89E-03 (6.32E-03) |
| Wind speed (m $s^{-1}$) | 3.89 (3.44) | 3.92 (3.65) | 4.46 (4.2) | 4.74 (4.44) |
| Temperature (°C) | 18.4 (15) | 12.7 (13.1) | 15.5 (14.6) | 17 (16.4) |
| $SO_2$ ($\mu g\ m^{-3}$) | 1.68 (2.23) | 1.33 (1.89) | 1.73 (1.75) | 1.74 (1.72) |
| $NO_x$ ($\mu g\ m^{-3}$) | 33.5 (55) | 28.5 (39.2) | 30.3 (39.4) | 24 (34.9) |
| $SO_4^{2-}$ ($\mu g\ m^{-3}$) | 1.93 (2.23) | 0.95 (1.36) | 0.98 (1.13) | 1.30 (1.47) |
| Particulate OC ($\mu g\ m^{-3}$) | 3.84 (4.90) | 2.24 (2.95) | 2.81 (2.96) | 2.43 (3.03) |

| Marylebone | | | | |
|---|---|---|---|---|
| | Continental | Arctic | Polar | Tropical |
| Condensation sink ($s^{-1}$) | 1.65E-02 (1.96E-02) | 1.16E-02 (1.57E-02) | 1.4E-02 (2.14E-02) | 1.82E-02 (2.39E-02) |
| Wind speed (m $s^{-1}$) | 3.92 (3.41) | 3.50 (3.64) | 3.84 (4.13) | 4.77 (4.4) |
| Temperature (°C) | 17.4 (15.2) | 13.4 (13.4) | 15.3 (14.8) | 16.9 (16.3) |
| $SO_2$ ($\mu g\ m^{-3}$) | 4.99 (6.39) | 4.31 (5.63) | 5.38 (7.43) | 6.95 (8.17) |
| $NO_x$ ($\mu g\ m^{-3}$) | 172 (250) | 139 (214) | 191 (303) | 269 (336) |
| $SO_4^{2-}$ ($\mu g\ m^{-3}$) | 3.24 (3.35) | 1.47 (1.6) | 1.52 (1.61) | 1.24 (1.8) |
| Particulate OC ($\mu g\ m^{-3}$) | 6.03 (6.91) | 3.81 (4.73) | 4.67 (5.97) | 5.31 (6.6) |

[Figure]

[Figure]

**Figure S1:** Example of a regional NPF event for all the sites of the present study (from top to bottom is HW, NK, MR)

Commented [DB(wISAS+2]: Addressing comment 1 (2nd)

[Figure]

[Figure]

Commented [DB(wISAS+3]: Addressing comment 8 (2nd)

[Figure]

[Figure]

[Figure]

[Figure]

[Figure]

[Figure]

[Figure]

**Figure S2:** Average and NPF event conditions for Harwell and N. Kensington and Marylebone Road. On Marylebone Road, Regional events' conditions refer to the Regional event days for all 3 sites.

[Figure]

[Figure]

[Figure]

[Figure]

[Figure]

[Figure]

[Figure]

[Figure]

[Figure]

[Figure]

**Figure S32:** Air mass origin frequency and conditions for NPF events in Harwell and N. Kensington.

[Figure]

**Figure S3:** Diurnal variation of $N_{20-50nm}$ for Harwell and N. Kensington during local and regional NPF events.

[Figure]

**Figure S44:** Weekly variation of NPF events in Marylebone Road.

[Figure]

[Figure]

**Figure S6:** Wind profile for Local (left) and Regional (right) NPF events in Marylebone Road.

---

## Author Response (AR2)

Journal: ACP
MS No.: acp-2018-1057
Title: Analysis of New Particle Formation (NPF) Events at Nearby Rural, Urban Background and Urban Roadside Sites
Author(s): Dimitrios Bousiotis et al.
MS Type: Research article
Iteration: Minor Revision

**RESPONSE TO THE CO-EDITOR**

**Comments to the Author:**
Dear authors,
Thanks for revising the manuscript.
The main concern of the referees is the large lowest detectable size of 16.6 nm of the SMPS system. The referees are right that it is difficult to assess the aerosol formation with the measurements starting with such a large size. As a result, you have modified the manuscript and softened some of the conclusions.
I still recommend to add a note about the measurement size also to the abstract and the conclusions.
**RESPONSE:** A note has been added in both the Abstract (line 31) and Conclusions (line 549) about the size range of the available data.

I have few additional comments that require some work.
Line 51: Clarify what do you mean by urban pollution.
**RESPONSE:** A clarification has been added about sources of urban pollution at the end of the abstract (line 51 of tracked version).

Line 125: What do you mean that particle formation was found to take place on event and non-event days.
**RESPONSE:** A clarification has been added about particle formation taking place but not qualifying as NPF events due to either newly formed particles not surviving or lack of growth (line 126).

Line 183: Confidence level typically implies statistical methods. Please modify the sentence.
**RESPONSE:** Confidence level has been changed to level of certainty (line 187)

Line 188: how frequent were the bursts detected with the CPC?
**RESPONSE:** An estimation of the frequency of the bursts and an explanation has been added (line 193). This estimation is based on a quick review of the data. To provide a more precise answer the whole dataset would need to be analysed again.

Line 223: What would the result be, if you would follow the mode through the night? Was growth typically persistent through the night? Setting a deadline for the midnight is rather arbitrary.
**RESPONSE:** A justification has been added of the reason why the end of the day was chosen as the final point for the growth rate calculation (line 228).

Line 237: NSF was proposed by who? Please add a reference.
**RESPONSE:** Reference added (line 245).

Line 280: From where was ammonia data available from? Please add a short section on ancillary data to the methods section.
**RESPONSE:** The data source has been added (line 166).

Line 299: From where did you get the organic compounds? VOC data it seems, but this is not described. Please describe in the methods section.
**RESPONSE:** The data source and measurement method has been added (line 168).

Line 315: Cluster 3 and following discussion is difficult to follow, but it clear after reading the section 3.3. Please summarise the trajectory analysis results here or consider structural changes.
**RESPONSE:** The chapter with the back trajectory analysis for the NPF events at the background sites was moved after the general back trajectory analysis for all three sites. Text and figure numbering were updated to match the changes made.

Line 320: again organic carbon concentration. From where?
**RESPONSE:** The data source has been added (line 169). Also, the text has been updated to indicate that the organic carbon concentration refers to both sites (line 487).

Line 323: low growth rate and consequently low survivability
**RESPONSE:** The text has been updated to address the suggested correction (line 490).

Line 329: How did you determine particulate organic carbon concentration?
**RESPONSE:** A data source has been added (line 167).

line 341: With the instrument that detects > 16.6. nm size distribution, one cannot assess the initial states of newly formed particles.
**RESPONSE:** The text has been updated to address the correction. "Initial stages" was changed to "early stages" (line 347).

Line 358: Ethane, from where is the data from?
**RESPONSE:** A data source has been added (line 167).

Table 1: A fraction of NPF days to all days would help to address the frequency at different locations.
**RESPONSE:** The number of days available per year were added in parentheses next to the number of events.

Table 2: What is the variability of these numbers?
**RESPONSE:** The table has been updated to present the variability of the values.

Figure 3: What is the variability of GR?
**RESPONSE:** The figure has been updated to present the variability of the values.

Figure 4: Mean or median? If the latter, quartile range would help to address the variability.
**RESPONSE:** The text of the figure has been updated to explain what is presented. The word "average" was replaced by the word "mean".

Figure 6: Same comment as Figure 3.
**RESPONSE:** The figure has been updated to present the variability of the values.

Figure 7: same as above.
**RESPONSE:** The figure has been updated to present the variability of the values.

[revised manuscript text omitted]

---

## Author Response (AR3)

| | |
|---|---|
| **acp-2018-1057**    Submitted on 03 Oct 2018
**Analysis of New Particle Formation (NPF) Events at Nearby Rural, Urban Background and Urban Roadside Sites**
Dimitrios Bousiotis, Manuel Dall'Osto, David C. S. Beddows, Francis D. Pope, and Roy M. Harrison | |
| First Contact: Roy M. Harrison, r.m.harrison@bham.ac.uk
Second Contact: D. C. S. Beddows, d.c.beddows@bham.ac.uk | |
| Agreed licence: Creative Commons Attribution 4.0 International | |
| **APC-discount application**
Executive Editor: application for 50% discount.
**Justification:** The University provides no direct support of open access charges, unless the work is funded by a UK research council. This work is funded only from the fee income of graduate student Dimitrios Bousiotis which is provided by his family. | |
| Handling Co-Editor: Tuukka Petäjä, tuukka.petaja@helsinki.fi
Manuscript Type: Research article | |
| **Status: File Upload (ACP)    Iteration: Correction** | |

**Minor Revision**

**Co-Editor Decision: Publish subject to technical corrections** (29 Mar 2019) by Tuukka Petäjä
Comments to the Author:
Dear Authors,

thanks for a quick and very good revision of the manuscript.

Only few technical issues remain, which I trust that you will correct accordingly.

Please spell out the acronyms VOC and OC in the data sources as they appear there for the first time. Furthermore, spell out DEFRA to make it clear for the reader that it connects to UK Air, Department of Environment, Food and Rural Affairs.

Please check, if you need to add a text in the acknowledgement section.

-tuukka

**RESPONSE**

Acronym DEFRA spelled at line 158

Acronym VOC spelled at line 167

Acronym OC spelled at line 168